# REVISITING THE EXPRESSIVENESS OF CNNs: A MATHEMATICAL FRAMEWORK FOR FEATURE EXTRACTION

## ABSTRACT

Over the past decade deep learning has revolutionized the field of computer vision, with convolutional neural network models proving to be very effective for image classification benchmarks. Given their widespread adoption, several theoretical works have analyzed their expressiveness, and study the class of piecewise linear functions that they can realize. However, a fundamental theoretical questions remain answered: why are piecewise linear functions effective for feature extraction tasks that arise in image classification? We address this question in this paper by introducing a simplified mathematical model for feature extraction, based on classical template matching algorithms that are commonly used in computer vision. We then prove that convolutional neural network classifiers can solve this class of image classification problems, by constructing piecewise linear functions that detect the presence of features, and showing that they can be realized by convolutional neurons. We also discuss the interpretability of the networks we construct, and compare them with those obtained via gradient-based optimization methods by conducting experiments on simple datasets.

## 1 INTRODUCTION

Over the past decade, convolutional neural network architectures have led to breakthroughs in a range of computer vision tasks, including image classification (25), object detection and semantic segmentation (39). Architectures such as AlexNet (18), VGG (40) and ResNet (28) have empirically shown that large convolutional neural networks can perform well on complex benchmarks such as ImageNet (12). While there has been theoretical research attempting to explaining their success in this context, fully understanding the underlying principles that contribute to their widespread adoption and effectiveness remains an open question.

Since their inception in the 1990s, it has also been empirically established that convolutional networks excel at feature extraction tasks (25). One of the major limitations they face is their lack of interpretability (46; 14); it difficult to pinpoint which features are being captured in their hidden layers. Advances in interpretability could also lead to the design of architectures that are both more transparent and efficient, by identify redundant parameters and thereby reducing the model's size and latency. Theoretical results can provide a deeper understanding of how these networks represent features internally and guide the development of models that are more interpretable.

In this work, we build on previous theoretical work studying the expressiveness of convolutional networks with ReLu activations, and examine the class of functions they can approximate. It is well-known that they compute piecewise linear functions, and the complexity of these function classes has been studied extensively (45), (33), (13), (34). A key result in this field is the universal approximation theorem, which shows that they can approximate piecewise linear functions with arbitrary precision (10; 29; 48). However, it is not clear if piecewise linear functions on the space of all input images can used for feature extraction problems, limiting the applicability of universal approximation theorems in computer vision.

By answering the mathematical question of why piecewise linear functions are effective for feature extraction, our key results bridge the gap between existing theoretical work analyzing the expressiveness of neural networks using these functions, and empirical results on the success of convolutional neural networks for computer vision problems. Our construction also provides insight into the interpretability of convolutional networks, and illustrates the principle of hierarchical compositionality

(13; 34) whereby simpler features are learnt in the earlier layers and used to construct higher-level features in the latter layers.

**Our contributions.** To address these problems, in this paper we present a rigorous mathematical framework that serves as a simplified model for image classification problems, based on classical template matching algorithms (6; 26; 17). Here rectangular regions of an image are compared to a predefined template, by sliding the template across the image and measuring the similarity at each position using the L1 distance metric. This framework, which is depicted below, serves as a simplistic model for the feature extraction problem in computer vision.

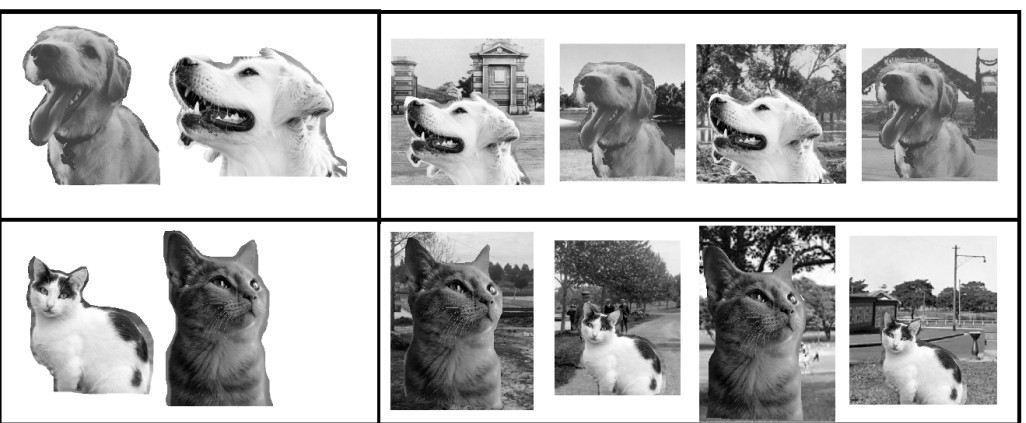

Figure 1: Features for "cat" and "dog"

Our key result constructs convolutional networks that can solve these image classification tasks. In the proof, we use piecewise linear functions that detect whether a given feature is present within a fixed rectangular region, and show that these functions can be realized by convolutional neurons. These functions, indexed by a feature and given region, are computed by the hidden neurons in the network, and output a non-zero value only if the region contains the feature. Our construction uses hidden neurons that are interpretable, with features learnt in a hierarchical fashion; lower-level (resp. higher-level) features are encoded in earlier (resp. latter) convolutional layers.

The paper is organized as follows. In Section 2, we formalize the feature extraction problem with convolutional neural networks and sketch the main results. In Section 3, we present our mathematical model for image classification, with examples illustrating the template matching procedure. In Section 4, we present our main results showing that convolutional networks can realize this function class corresponding to image classification tasks. In Section 5, we outline the proof, focusing on the construction of piecewise linear functions that extract features. In Section 6, we perform an experimental analysis using features extracted from Fashion-MNIST, to compare the piecewise linear functions we construct with those obtained via gradient-based optimization methods. In the Appendix, we provide detailed proofs of the main theoretical results.

**Related work.**

*Approximation theory.* In the 1990s, it was shown that a neural network with a single hidden layer can approximate any continuous function provided that its width is sufficiently large (10), (29). More recently, analogous results were established for deep neural networks with ReLu activations (21; 24; 31), and similar results were established for deep convolutional neural networks in (48). While these results give valuable insights, they do not explain why the class of piecewise linear functions is suitable for feature extraction, limiting their applicability to computer vision questions. Our work bridges the gap by answering this question using a simplified model based on classical template matching algorithms for vision, and serves as a starting point for a more comprehensive theory explaining the real-world success of convolutional networks.

*Expressiveness of neural networks.* Another line of work analyzes the number of linear regions in the piecewise linear function that is computed by a neural network with ReLu activations (33), (36).

Using combinatorial results, they derive lower and upper bounds for the maximal number of linear regions in a fully connected ReLu network with $L$ hidden layers and pre-specified widths (41), (4), (22; 23), (47). These results show that deep fully connected networks can express functions with exponentially more linear regions than their shallower counterparts (35), (42). While analogous results for deep convolutional networks were established recently (45), it is unclear how this relates to real-world vision datasets. Our work complements the above paper, by introducing a class of piecewise linear functions that excel at feature extraction which are crucial to the empirical success of convolutional networks.

*Hierarchical compositionality.* Hierarchical compositionality in computer vision refers to the principle of constructing complex images from smaller features in a layered fashion (13; 34; 1). In recent work (8), the authors analyze this phenomena empirically using synthetic data generated from a random hierarchy model. In (32), a similar generative model for vision data is presented, and used to analyze the convergence of a clustering algorithm using stochastic gradient descent. By using a template matching framework for feature extraction, our work provides additional insight by explicitly constructing CNNs where the neurons in earlier layers of the network recognize simpler features (eg. $2 \times 2$ patches) which are then used to recognize more complex features in latter layers.

*Interpretability.* A key obstacle to the interpretability of convolutional networks is the principle of superposition, where multiple features are encoded simultaneously within a single neural dimension, making individual feature disentanglement challenging (14). In (46), a loss function is added to train networks where filters in the last convolutional layer correspond to a specific class. Sparse autoencoders (11; 30; 5) have been used to combine multiple neuron activations in the original network into a neuron that detects a specific feature. While a full explanation of how the hidden layers of a convolutional network function remains elusive, our work introduces a simplified mathematical model that can be used to study these phenomena through the lens of piecewise linear functions.

## 2 PRELIMINARIES

### 2.1 IMAGE CLASSIFICATION

In this section we introduce notation for convolutional neural networks and image classification tasks that will be used throughout the paper. We represent the input image using a rectangular matrix whose entries are scaled so their values between $0$ and $1$. For simplicity we only consider black-and-white images, noting that it is straightforward to extend our results to the multi-channel setting with color images. The value of an entry in the matrix represents the shade of grey present in the corresponding portion of the image. We then formalize the image classification problem below, using a pre-specified set of image labels $\mathcal{L}$ (such as "cat", "dog", etc). The objective is to construct an image classification map with zero error.

**Definition 2.1.** Denote the input space as follows.
$$\mathcal{X}_{m,n} = \{\underline{x} = (x_{i,j}) \in \mathrm{Mat}_{m \times n}(\mathbb{R}) \mid 0 \leq x_{i,j} \leq 1\} \qquad \blacksquare$$

**Definition 2.2.** Let $\mathcal{L}$ be the finite set consisting of all image labels. For each image label $l \in \mathcal{L}$, let $\mathcal{X}_{m,n}^l \subset \mathcal{X}_{m,n}$ denote the set of all input matrices that contain the image corresponding to $l$, and no other images. Denote by $\mathcal{X}_{m,n}^{\mathcal{L}}$ the set of all image matrices containing one of the images in $\mathcal{L}$. An *image classification map* is a function $f : \mathcal{X}_{m,n}^{\mathcal{L}} \to \mathcal{L}$. We say that the map $f$ has zero error if the following holds.
$$\mathcal{X}_{m,n}^{\mathcal{L}} = \bigsqcup_{l \in \mathcal{L}} \mathcal{X}_{m,n}^l; \qquad \underline{x} \in \mathcal{X}_{m,n}^l \Rightarrow f(\underline{x}) = l \qquad \blacksquare$$

We are specifically interested in image classification maps using convolutional neural networks, which we define below. We refer the reader to Appendix A for the basic definitions of convolutional/fully connected layers, and ReLu activation functions.

**Definition 2.3.** A convolutional neural network $\mathcal{N}$ consists of $L$ convolutional layers, a flattening layer, and $L'$ fully connected layers (typically $L'$ is 1 or 2). Denote the weights and biases of the $i$-th convolutional layer by $(\underline{w}_i, \underline{b}_i)$, and those of the $i$-th fully connected layer by $W_i = (A_i, B_i)$. The induced function $f_{\mathcal{N}} : \mathrm{Mat}_{m,n}(\mathbb{R}) \to \mathbb{R}^l$ is as follows, where $l$ denotes the output dimension.
$$f_{\mathcal{N}}(\underline{x}) = \overline{\phi}_{W_{L'}} \circ \cdots \circ \overline{\phi}_{W_1} \circ \phi_{fl} \circ \overline{\phi^c}_{(\underline{w}_L, \underline{b}_L)} \circ \cdots \circ \overline{\phi^c}_{(\underline{w}_1, \underline{b}_1)}(\underline{x}) = (f_{\mathcal{N}}^1(\underline{x}), \cdots, f_{\mathcal{N}}^l(\underline{x}))$$

We denote by $\overline{f_\mathcal{N}}$ the classification function corresponding to the convolutional neural network $\mathcal{N}$.

$$\overline{f_\mathcal{N}} : \mathrm{Mat}_{m,n}(\mathbb{R}) \to [1, 2, \cdots, l]; \qquad \overline{f_\mathcal{N}}(\underline{x}) = \operatorname*{argmax}_{1 \leq i \leq l} f_\mathcal{N}^i(\underline{x}) \qquad \blacksquare$$

## 2.2 OVERVIEW OF MAIN RESULTS.

We now give a sketch of the main results in this paper. While the classification map $f$ above cannot be the restriction of a continuous function on $\mathcal{X}_{m,n}$, in the context of convolutional networks it is obtained by taking the maximum of the $l$ output logits, which are themselves piecewise linear functions. This leads us to the question of whether there exist continuous functions corresponding to the output logits, which we denote by $\{f_l(x)\}_{l \in \mathcal{L}}$, with the following feature extraction property.

$$\underline{x} \in \mathcal{X}_{m,n}^l \Rightarrow f_l(\underline{x}) > f_{l'}(\underline{x}) \text{ if } l' \neq l$$

Our first key contribution is to formally define a simplified mathematical model for feature extraction (in particular, the image classes $\mathcal{X}_{m,n}^l$) and establish the existence of piecewise linear functions with the above property. We then construct a convolutional network classifier $f : \mathcal{X}_{m,n}^\mathcal{L} \to \mathcal{L}$ which has zero errors, highlighting the interpretability of the classifier and how the functions of the hidden neurons can be understood through the principle of hierarchical compositionality. The details of these constructions will be presented in the next three sections.

## 3 A MATHEMATICAL FRAMEWORK FOR IMAGE CLASSIFICATION

In this section, we present a rigorous mathematical framework for image classification problems. Our framework is closely related to template matching, a classical technique in computer vision that locates parts of an image matching a predefined template (6; 26; 17). It works by sliding the template across the image and computing a similarity measure to see whether or not there is a match. By using this simplified framework with features being defined by predefined templates, in the subsequent sections we will illustrate how piecewise linear functions on the input space can be used to extract features, which in turn leads to our results on the expressiveness of CNNs in this context.

### 3.1 IMAGE CLASSIFICATION

We start with the observation that an image consists of a set of features that define it (13), (34), (43). For simplicity, our model stipulates that each feature can be characterized by a finite collection of fixed templates. For examples, a mouth would be defined by a set of distinct templates, each of which resembles a human mouth. We introduce the notion of a "feature tile" to describe the constituent templates.

**Definition 3.1.** Given a matrix $m \in \mathrm{Mat}_{m,n}(\mathbb{R})$, define its *support* $\mathrm{supp}(m)$ as follows.

$$\mathrm{supp}(m) = \{(i,j) \mid 1 \leq i \leq m, 1 \leq j \leq n; m_{i,j} \neq 0\} \qquad \blacksquare$$

**Definition 3.2.** A **feature tile** $T$ with dimension $k \times l$ is a pair $T = (t, \epsilon)$ with $t \in \mathcal{X}_{k,l}$ and $\epsilon > 0$. $\blacksquare$

**Definition 3.3.** Given a feature tile $T = (t, \epsilon)$ with dimension $k \times l$ and an image $x \in \mathcal{X}_{k,l}$ define the quantity $\underline{t}(x)$ as follows.

$$\underline{t}(x) = \sum_{(i,j) \in \mathrm{supp}(t)} |x_{i,j} - t_{i,j}| \qquad \blacksquare$$

The quantity $\underline{t}(x)$ is used to determine whether or not the image $x$ contains the tile $T$, with the parameter $\epsilon$ bounding the discrepancy between the two. The sum is taken over $\mathrm{supp}(t)$ in the case where the feature that $T$ contains is not a full rectangle, but rather a subset of pixels inside a rectangle (i.e. the non-zero coordinates of $t$ contain the relevant feature). Below, we define the space of images $\mathcal{X}_{m,n}^T$ containing the feature tile $T$ via the sliding window approach used in template matching algorithms; the subscripts $[i+1, i+k] \times [j+1, j+l]$ specify the region of the input image that contains the feature tile.

**Definition 3.4.** Given a feature tile $T = (t, \epsilon)$ with dimension $k \times l$, define $\mathcal{X}^T \subset \mathcal{X}_{k,l}$ and $\mathcal{X}^T_{m,n} \subset \mathcal{X}_{m,n}$ as follows. Given $\underline{x} \in \mathcal{X}_{m,n}$, below $\underline{x}_{[i+1,i+m],[j+1,j+n]}$ denotes the sub-matrix with rows indexed by $[i+1, \cdots, i+m]$ and columns indexed by $[j+1, \cdots, j+n]$.

$$\mathcal{X}^T = \{x \in \mathcal{X}_{k,l} \mid \underline{t}(x) \leq \epsilon\}$$
$$\mathcal{X}^T_{m,n} = \{\underline{x} \in \mathcal{X}_{m,n} \mid \exists \, i, j \text{ such that } \underline{x}_{[i+1,i+k],[j+1,j+l]} \in \mathcal{X}^T\} \qquad \blacksquare$$

We say that an input matrix contains an image $\mathcal{I}$ if it contains any of the constituent feature tiles corresponding to that image, so the corresponding subset of $\mathcal{X}_{m,n}$ is given by taking the union. Our image classification problem will be defined by a set of images. To avoid ambiguity, the corresponding set of image matrices will exclude any which contain multiple images.

**Definition 3.5.** An **image** $\mathcal{I}$ is a set of **feature tiles**, $\mathcal{I} = \{T_1, T_2, \cdots, T_q\}$, with $T_i = (t_i, \epsilon_i)$ for $1 \leq i \leq q$. Note that the dimensions of the tiles $T_i$ need not be the same. Define $\mathcal{X}^{\mathcal{I}}_{m,n}$, the space of all images $\mathcal{X}_{m,n}$ containing the image $\mathcal{I}$, as follows.

$$\mathcal{X}^{\mathcal{I}}_{m,n} := \bigsqcup_{i=1}^{q} \mathcal{X}^{T_i}_{m,n}$$

**Definition 3.6.** An **image class** $\overline{\mathcal{I}}$ consists of a set of images $\overline{\mathcal{I}} = \{\mathcal{I}_1, \cdots, \mathcal{I}_l\}$. Define $\mathcal{X}^{\overline{\mathcal{I}}}_{m,n}$ to be the set of images which corresponds to exactly one of the images in $\overline{\mathcal{I}}$.

$$\mathcal{X}^{\overline{\mathcal{I}}}_{m,n} = \bigcup_{j=1}^{l} \mathcal{X}^{\mathcal{I}_j}_{m,n} - \bigcup_{1 \leq j < j' \leq l} \mathcal{X}^{\mathcal{I}_j}_{m,n} \cap \mathcal{X}^{\mathcal{I}_{j'}}_{m,n} \qquad \blacksquare$$

In Figure 1 (see the introduction), we use the above framework to model a real-world image classification task, with two labels: "cat" and "dog". For each image, in the left we have two feature tiles of differing dimensions corresponding to the object, which are non-rectangular pictures extracted from real-world images. On the right there are four examples, which are generated by superimposing one of the feature tiles above various background images; note that our framework does not impose any restrictions on the background image. These resulting images are very realistic, and show that our framework based on template matching effectively models the feature extraction problem.

# 4 MAIN RESULTS

Our main results in this section solve the image classification problem presented in Section 3, and construct a convolutional network classifier that accurately predicts the labels of input matrices from $\mathcal{X}^{\overline{\mathcal{I}}}_{m,n}$. Existing work on expressiveness of deep networks focuses primarily on the setting whether the functions being approximated are continuous; while this covers a large class of regression problems, it does not immediately extend to discrete image classification functions that cannot immediately be expressed using continuous (or piecewise linear) functions. To bridge this gap, we start by constructing piecewise linear functions that are effective for feature extraction, and then show these functions can be effectively realized by convolutional networks. See Appendix B for detailed proofs.

## 4.1 ARE PIECEWISE LINEAR FUNCTIONS EFFECTIVE FOR FEATURE EXTRACTION?

The output logits of convolutional neural networks compute piecewise linear functions, which leads us to the question of how they can separate different image classes. The first key step in the proof is to construct a piecewise linear function on $\mathcal{X}_{m,n}$, the space of all input images, that detects whether or not an input matrix contains a given feature. The value of the function should be non-zero if the input image contains the feature, and zero otherwise. We formulate this precisely below; the sliding window technique used for template matching is the key to constructing these functions.

**Proposition 1**. Given an image $\mathcal{I} \in \overline{\mathcal{I}}$, there exists a piecewise linear function $\phi_{\mathcal{I}}(\underline{x})$ such that the following statement holds.

$$\phi_{\mathcal{I}}(\underline{x}) : \mathcal{X}_{m,n} \to \mathbb{R}$$
$$\phi_{\mathcal{I}}(\underline{x}) > 0 \Leftrightarrow \underline{x} \in \mathcal{X}^{\mathcal{I}}_{m,n} \qquad \blacksquare$$

## 4.2 ON THE EXPRESSIVENESS OF CNNs.

We state our first key result below. Using the above proposition, the next step is to show that the piecewise linear functions can be realized as the output logits of a convolutional network. Once the weights of the network have been chosen so that this holds, for a given input image the output logit corresponding to the feature it contains will be non-zero, and the other output logits will be zero, yielding a classifier with zero error. The hidden neurons in the fully connected layer are interpretable, and detect whether a given feature is present within a fixed rectangular region of the input image.

**Theorem 1**. Let $\overline{\mathcal{I}} = \{\mathcal{I}_1, \cdots, \mathcal{I}_l\}$ be an image class. There exists a network $\mathcal{N}[\overline{\mathcal{I}}]$ with one convolutional layer with $2 \times 2$ kernels and one fully connected layer such that the induced classifier $\overline{f}_{\mathcal{N}[\overline{\mathcal{I}}]} : \mathcal{X}_{m,n}^{\overline{\mathcal{I}}} \to \overline{\mathcal{I}}$ has zero error.

## 4.3 DEEP CNNs AND HIERARCHICAL COMPOSITIONALITY

It is widely understood that the success of deep convolutional networks is due to the principle of hierarchical compositionality (see (13; 34)), whereby complex structures are obtained by combining simpler ones in a hierarchical fashion. Deeper layers in the network can recognize more complex features, building upon the simpler features that were detected by earlier layers. Theorem 2 below, which is a variant of Theorem 1, highlights this concept. In the proof, we interpret the hidden neurons in the convolutional layers, and show how neurons in earlier (resp. latter) layers can detect smaller (resp. larger) patches of feature tiles.

**Theorem 2**. Let $\overline{\mathcal{I}} = \{\mathcal{I}_1, \cdots, \mathcal{I}_l\}$ be an image class, such that each image $\mathcal{I}_j$ has size less than $2^r$ for some $r$. There exists a network $\mathcal{C}[\overline{\mathcal{I}}]$ with a padding layer, $r$ convolutional layers and one fully connected layers such that the induced classifier $\overline{f}_{\mathcal{C}[\overline{\mathcal{I}}]} : \mathcal{X}_{m,n}^{\overline{\mathcal{I}}} \to \overline{\mathcal{I}}$ has zero error.

## 5 PROOFS

### 5.1 PROOF OF PROPOSITION 1: WHY ARE PIECEWISE LINEAR FUNCTIONS EFFECTIVE FOR FEATURE EXTRACTION?

The feature tile is the fundamental building block of an image class. To construct $\phi_{\mathcal{I}}(\underline{x})$ from Proposition 1, we start by defining a piecewise linear function $\phi_T(\underline{x})$ for a given feature tile $T$. Motivated by the algorithm for template matching, we slide the feature tile across all rectangular patches (indexed by $\mathcal{R}_{k,l}^{m;n}$ below), use the similarity function $\underline{t}$ to see if the feature tile appears in each patch, and take the sum over all rectangular patches.

**Definition 5.1.** We define $\mathcal{R}_{k,l}^{m,n}$, which indexes all sub-rectangles of size $k \times l$ inside a larger rectangle of size $m \times n$ as follows. Given a feature tile $T = (t, \epsilon)$ and an image $\underline{x} \in \mathcal{X}_{m,n}$, define $\phi_T(x)$ as follows.

$$\mathcal{R}_{k,l}^{m,n} = \{(i,j) \mid i+k \le m, j+l \le n\}; \qquad \phi_T(\underline{x}) = \sum_{(i,j) \in \mathcal{R}_{k,l}^{m,n}} \max(0, \epsilon - \underline{t}(\underline{x}_{[i+1,i+k],[j+1,j+l]}))$$

**Lemma 5.1.** The following inequality holds: $\phi_T(\underline{x}) > 0 \Leftrightarrow \underline{x} \in \mathcal{X}_{m,n}^T$

The above Lemma can be deduced from Definition 3.4. Using this Lemma, we can define the piecewise linear function $\phi_{\mathcal{I}}(\underline{x})$ below by simply taking the sum of the corresponding functions for the constitutent feature tiles. Using Definition 3.5, we can deduce that it has the desired property from Proposition 1. We refer the reader to Appendix B.1 for all proofs in this subsection.

**Definition 5.2.** Given an image $\mathcal{I} = \{T_1, \cdots, T_q\}$ and an input matrix $\underline{x} \in \mathcal{X}_{m,n}$, define $\phi_{\mathcal{I}}(\underline{x})$ as follows.

$$\phi_{\mathcal{I}}(\underline{x}) = \sum_{1 \le i \le q} \phi_{T_i}(\underline{x}) \qquad \blacksquare$$

### 5.2 PROOF OF THEOREM 1: AN INTERPRETABLE SINGLE-LAYER CNN

In order to prove Theorem 1 using Proposition 1, it remains to construct convolutional neural networks whose output logits are the piecewise linear functions $\phi_{\mathcal{I}_j}(\underline{x})$. Since they are obtained by summing

the corresponding functions $\phi_T(\underline{x})$ where $T$ is a feature tile, it suffices to realize the latter functions using a convolutional network. We formalize this in the below Lemma; see Appendix B.2 for complete proofs in this subsection.

**Lemma 5.2.** Let $T = (t, \epsilon)$ be a feature tile with dimension $k \times l$. There exists a convolutional neural network $\mathcal{N}[T]$ with one hidden convolutional layer and one hidden fully connected layer such that the following holds: $f_{\mathcal{N}[T]}(\underline{x}) = \phi_T(\underline{x})$

In order to construct the function $\phi_T(\underline{x})$ with a convolutional network, we first need to construct the functions $\underline{t}(\underline{x}_{[i+1,i+k],[j+1,j+l]})$ that are used in its definition. The key observation is that they can be realized using convolutional filters with ReLu activations via the below identity.

$$||y - c|| = max(y - c, c - y) = \sigma(2y - 2c) - \sigma(y) + c \qquad \text{for } y, c \in \mathbb{R}$$

A key feature of our construction is that the hidden neurons in the fully connected layers realize the functions $t(\underline{x}_{[i+1,i+k],[j+1,j+l]})$, which detect whether the corresponding rectangle contains the feature $T$. This implies means the hidden neurons in the fully connected layer are interpretable, and have sparse activations. For a given image, only the hidden neuron corresponding to the rectangle region containing the feature in question will activate, and the others will be zero.

### 5.3 PROOF OF THEOREM 2: INTERPRETING DEEP CNNs VIA HIERARCHICAL COMPOSITIONALITY

In the proof of Theorem 2 (see Appendix B.3 for details), the key difference is that instead of using the hidden fully connected layer to realize the functions $\underline{t}(\underline{x}_{[i+1,i+k],[j+1,j+l]})$, we construct them in the hidden convolutional layers using the principle of hierarchical compositionality. We define the matrices $D_{\underline{t}}(\underline{x})$ below, based on the sliding window technique used in template matching, and then proceed to construct them inductively with a deep convolutional neural network. As in the proof of Theorem 1, the function $\phi_T(\underline{x})$ can be constructed from the entries of $D_{\underline{t}}(\underline{x})$, yielding the desired classifier.

**Definition 5.3.** Given a matrix $t \in \text{Mat}_{p,q}(\mathbb{R})$ and $\underline{x} \in \mathcal{X}_{m,n}$, define $D_{\underline{t}}(\underline{x}) \in \text{Mat}_{m-p+1,n-q+1}(\mathbb{R})$ as follows. Here we using the notation from Definition 3.3, and assume $p < m$ and $q < n$.

$$D_{\underline{t}}(\underline{x}) = \begin{pmatrix} \underline{t}(\underline{x}_{1:p,1:q}) & \underline{t}(\underline{x}_{1:p,2:q+1}) & \cdots \\ \underline{t}(\underline{x}_{2:p+1,1:q}) & \cdots & \cdots \\ \cdots & \cdots & \cdots \end{pmatrix}$$

**Lemma 5.3.** Let $T = (t, \epsilon)$ be a feature tile with dimension $k \times k$, where $k = 2^r$ for some $r \geq 1$, and let $\underline{x} \in \mathcal{X}_{m,n}$. There exists a convolutional neural network $\mathcal{N}[t]$ with $r + 1$ convolutional layers such that the following holds. The $(i + 1)$-st convolutional layer of $\mathcal{N}[T]$ has $2^i \times 2^i$ kernels.

$$f_{\mathcal{N}[t]}(\underline{x}) = D_t(\underline{x}) \qquad \blacksquare$$

A key feature of our construction is that the hidden neurons in the convolutional layers can be interpreted as follows (note that we pad our feature tiles so they have dimension $2^r \times 2^r$ for some $r$). We can divide the feature tile $T$ into rectangular patches $t'$ of size $2^i \times 2^i$; then each neuron in the $i$-th convolutional layer computes a function $t'(\underline{x}_{[r+1,r+2^i],[s+1,s+2^i]})$, which detects whether the corresponding rectangle contains the patch $t'$. This illustrates hierarchical compositionality, as earlier layers of the convolutional network detect whether rectangular regions contain smaller patches (eg. of size $2 \times 2$), and the latter layers correspond to larger patches of $T$.

## 6 EXPERIMENTAL RESULTS

In this section, we conduct experiments comparing the networks from our theoretical frameworks with those obtained via training with stochastic gradient descent. First we conduct experiments on a toy dataset with black and white squares to investigate the properties of convolutional neurons, and then study similar questions with convolutional networks trained on the Fashion-MNIST dataset (44; 27). All experiments are conducted using PyTorch; see the supplementary sections for the code.

### 6.1 A toy experiment: why do CNNs distinguish black and white squares?

We start with a toy model, where the convolutional network is trained to classify $4 \times 4$ greyscale images. We use a simple example of our framework where there are two classes, which we will denote by $\mathcal{I}_1$ and $\mathcal{I}_2$, each of which contain a single feature tile with dimension $2 \times 2$ - the white square is a matrix with four zeroes, and the black square is a matrix with four ones. This toy dataset is illustrated in the below figure, where each image contains a $2 \times 2$ square that is either white or black.

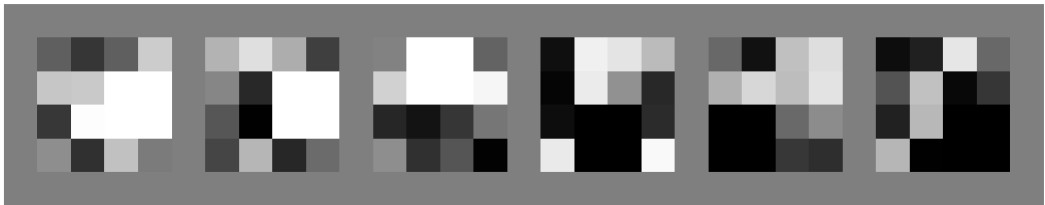

Figure 2: Dataset samples showcasing black and white squares on random backgrounds

We train a convolutional network on this dataset, using a convolutional layer with a single $2 \times 2$ filter, followed by a fully connected layer with dimension $2 \times 9$ (note that the output of the convolutional layer has dimension $3 \times 3$, and has 9 neurons after flattening). After training on a labelled dataset with 400 samples using stochastic gradient descent with a learning rate of 0.05 for 100 epochs, the convolutional network reaches an accuracy of 95%. While this can be improved to over 99% by increasing the number of filters, for simplicity we use this setup to explain the inner workings of this network. The weights and biases of both layers are below.

$$\texttt{conv1.weight} = \begin{bmatrix} 1.02 & 1.04 \\ 1.32 & 0.76 \end{bmatrix}, \texttt{conv1.bias} = -1.65, \texttt{fc1.bias} = \begin{bmatrix} -1.97 & 2.60 \end{bmatrix}$$

$$\texttt{fc1.weight} = \begin{bmatrix} 0.09 & 0.44 & 0.51 & 0.74 & 1.26 & 0.74 & -0.06 & 0.74 & 0.18 \\ -0.23 & -0.39 & -0.33 & -0.64 & -1.30 & -0.60 & -0.46 & -0.50 & -0.05 \end{bmatrix}$$

We now provide some quantitative heuristics based on these parameters explaining why this network performs well on this classification task. If the input image contains a square with four values whose average is at least 0.7 (resp. at most 0.4), the corresponding convolutional neuron will typically have a value greater than 1 (resp. 0). If the convolutional layer has an average value greater than 0.5 (resp. less than 0.3), the first output logit will typically have a positive (resp. negative) value. It follows that an image containing a black (resp. white) square will usually lead to the first output logit being positive (resp. negative), based on the value of the convolutional neurons corresponding to the region near the black/white square.

We note that the convolutional filter plays a similar role to those constructed theoretically in Theorem 1 as they help detect the white square feature - each neuron has value 0 if the corresponding $2 \times 2$ patch resembles a white square. However, the output logits here are not sparse, whereas in Theorem 1 an output logit has a zero value if the input image doesn't correspond to that logit. Analyzing larger networks trained on toy datasets similar to this one, and in particular studying the piecewise linear functions computed by the output logits, could lead to more insights into the interpretability of convolutional networks.

### 6.2 How do CNNs classify Fashion-MNIST with piecewise linear functions?

In this section, we conduct experiments using the image classification frameworks that we construct, with features extracted from Fashion-MNIST (44; 27). We generate image classes $\mathcal{I}$ as follows, with one class for each of the ten labels in Fashion-MNIST, each of which consists of $k$ feature tiles that are obtained by randomly choosing a sample from the training set of Fashion-MNIST with the corresponding label, and setting $t$ to be the $28 \times 28$ matrix obtained. To generate images from $\mathcal{X}^{\mathcal{I}}$, we start by randomly generating an grayscale image with dimension $40 \times 40$, randomly selecting a rectangular subpatch with dimension $28 \times 28$, and replacing it with one of the 10 feature tiles. This process is illustrated in the below figure, with feature tiles displayed in the first row and the image directly underneath lying in the corresponding image class $\mathcal{X}^{\mathcal{I}}$.

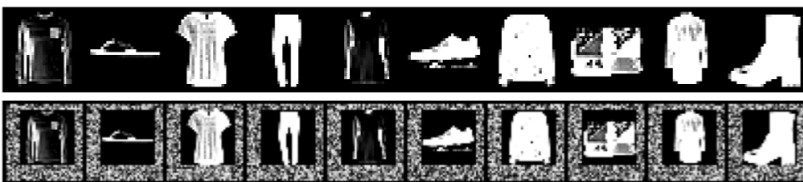

Figure 3: Dataset samples showcasing feature tiles from Fashion-MNIST on random backgrounds

We use a convolutional neural network architectures for this image classification task, which consists of a single convolutional layer followed by fully connected layer that yields the output logits. The convolutional layer has $5 \times 5$ kernels, and $12$ convolutional filters. The networks are trained using stochastic gradient descent for 15 epochs with a learning rate set to 0.03, batch size 100, and a cross-entropy loss function. Using a dataset with $4000$ training samples and $1000$ testing samples, the trained networks consistently reach an accuracy of over $99\%$.

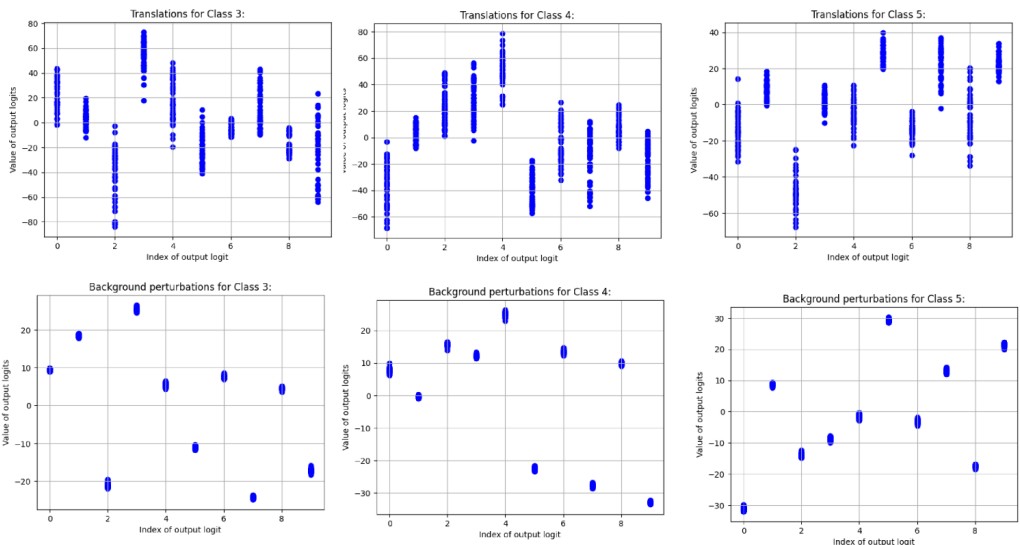

Figure 4: Output logits for randomly generated images obtained from three feature tiles.

We now analyze the piecewise linear functions computed by the output logits $\{f_N^i(x)\}$ in the network $N$ obtained empirically, and compare them with our theoretical results. Given a feature tile $T$ corresponding to a class $l$, and an image containing that feature, these piecewise linear functions should have the following property: regardless of what the background consists of, and regardless of where $T$ is positioned in the image, the index of the largest output logit is $l$. In other words, the maximum index of the output logits should be robust to *translations* and *background perturbations*.

To illustrate this, in the plot below we choose three feature tiles $T$ from different classes. In the first row, row the 100 generated images are obtained using background perturbations with the tile in a fixed position. In the second row, the generated images are obtained by translating the tile with a fixed background. The distribution of the output logits is plotted in both, and we find that their variance is quite small when we add background perturbations, but much larger when the tile is translated. In contrast, in our theoretical framework the piecewise linear functions we construct remain unchanged in the presence of background perturbations, or translations.

The key difference is that while our construction uses "monosemantic" neurons that are only activated by a single feature, as observed in (14) we find that gradient-based optimized yields networks with "polysemantic" neurons that fire for many unrelated features. To bridge the gap between our theoretical results and empirical observations, further work is needed to decompose the piecewise

linear functions in the output logits into units that can be interpreted. Our theoretical insights will be useful for analyzing these piecewise linear functions and understanding why they are robust to background perturbations, which is essential for interpreting the inner workings of convolutional networks.

## 7 DISCUSSION AND FURTHER DIRECTIONS

**Limitations of template matching.** Our mathematical framework for generating images based on template matching illustrates how piecewise linear functions can be used for feature extraction, and is a starting point for a theory explaining the success of convolutional networks. However, template matching algorithms (6; 26; 17) do not suffice to fully explain the real-world success of convolutional networks. While template matching can handle cases where a portion of image is occluded (19), it does not cover scenarios where the image is rotated, distorted, or its texture modified (16). In order to account for these scenarios, it would be interesting to generalize our template matching framework, which uses a discrete collection of feature tiles for each image, to a setting where features corresponds to a continuous spectrum of shapes (for instance, an eye should correspond to a spectrum of shapes and textures obtained by deforming a given eye, and not a finite set of eye shapes).

**Interpretability.** While the convolutional networks that we construct contain monosemantic neurons, which are only activated for images of a specific class, optimization methods yield networks with polysemantic neurons, which activate for multiple unrelated classes (14). Our experiments indicate that even for solving template matching problems with convolutional networks, bridging the gap between monosemanticity and polysemanticity remains an important open question. While there has been progress in this direction using sparse autoencoders (11; 30; 5), this does not fully explain the inner workings of convolutional networks. Our theoretical framework for extracting features using piecewise linear functions could be used to investigate how the output logits of trained networks perform feature extraction, and determine if there is a subset of neurons that are responsible for recognizing a given feature.

**Sparsity.** Empirically it has been observed that neural networks trained on computer vision datasets can be sparsified without a drop in accuracy (15), (20), but the theoretical underpinnings of this phenomena are not fully understood. Our simplified model for classification provides some insight into sparsity patterns; in particular our networks in Theorem 1 use a fully connected layer which contain weights that have values 0 or 1, whereby each convolutional filter is connected to a unique output logit with non-zero weight. While existing work on lottery tickets (15) yields pre-defined sparsity patterns that are difficult to interpret, generalizing our theoretical results in Theorem 2 could lead to new block sparsity patterns for convolutional layers with theoretical guarantees.

**Stochastic gradient descent.** While our work builds on existing results studying the expressiveness of convolutional networks, another line of work investigates their learning capabilities using gradient-based optimization, and establishes theoretical convergence guarantees with varying assumptions (2; 3; 7; 49). Existing work has answered the question of how these networks memorize data with gradient-based optimization, but this doesn't extend to feature extraction problems that arise in real-world applications. Our experimental results indicate that convolutional neural networks can achieve near-perfect accuracies on our synthetic datasets. It would be interesting to extend our results and obtain theoretical convergence guarantees, and determine whether overparametrization is necessary in this setting (2; 49).

## 8 CONCLUSION

In this paper, we present a novel mathematical framework based on template matching that can be used as a simplified model of real-world computer vision tasks. Our key insight is a construction of piecewise linear functions that effective for feature extraction, which we use to analyze the expressiveness of convolutional networks and show that they can solve image classification tasks. We do not anticipate any immediate negative societal impacts, as the present work is primarily theoretical in scope. We also discuss implications for the interpretability of convolutional networks, based on the principle of hierarchical compositionality. Our work is a starting point for a rigorous mathematical theory explaining the effectiveness of convolutional networks for computer vision, which is crucial for understanding the interpretability of these models.

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
