# OpenReview forum: "Revisiting the expressiveness of CNNs: a mathematical framework for feature extraction"
_ICLR.cc/2025/Conference — ICLR 2025 Conference Withdrawn Submission_

### Official Review · Reviewer_oRvL · 2024-10-26

**Soundness:** 2
**Presentation:** 2
**Contribution:** 1
**Rating:** 3
**Confidence:** 4

**Summary:**

The paper  introduces a mathematical model to analyze RELU convolutional neural networks (CNNs) for image classification.
The authors goal is to answer the following question that they say is a  fundamental theoretical question:  why are piecewise linear functions effective for feature extraction tasks that arise in image classification?  Their conjecture is that the networks may perform a hierarchical version of classical template matching. They then proceed to show, in an interesting way, how this can be done using piecewise linear functions. Finally they perform experiments to check how their idealized networks compare with those obtained via gradient-based optimization methods.

**Strengths:**

The original contribution of the paper is to show in detail how to construct RELU networks that perform hierarchical template matching. This is interesting and mathematically non-trivial.

**Weaknesses:**

My reading of the paper is that it proposes an hypothesis -- an extension of classical template matching -- of how CNNs using RELUs could perform image classification. Then the  hypothesis is used to build an interesting theoretical framework. Finally experiments with RELU networks trained in the usual way do not support the predictions.

A natural reaction may well be: why to publish a theory that is not supported by the experiments? I am more tolerant: I am not against publishing negative results and in fact I think that it is sometime a good idea. However, the fact that the theory is falsified by the experiments should be made clear throughout the paper, for instance  in the abstract. This may be an approach for how to rewrite the paper for a future submission.

There is another significant problem however. The authors say "a fundamental theoretical questions remain answered: why are piecewise linear functions effective for feature extraction tasks that arise in image classification?". I think the general answer to the  question has been pretty clear for a long time; it is well known that piecewise linear splines are universal approximators in d dimensions. The hierarchical version of this for deep NNs can be found in https://link.springer.com/article/10.1007/s11633-017-1054-2. Theorems by Yarotski are relevant for the case of strict RELUs, see https://www.sciencedirect.com/science/article/abs/pii/S0893608017301545.

Minor problems:
-- there are typos (example in the abstract: z"...a fundamental theoretical questions..."
-- meaningless sentences (example: "...they (piecewise linear functions) can approximate piecewise linear functions with arbitrary precisions..."

**Questions:**

As I mentioned, I think the question of why are piecewise linear functions effective for feature extraction tasks that arise in image classification is not fundamental and is, in any case, answered by approximation theory. Therefore it is not clear to me what you mean when  you say "...why the class of piecewise linear functions is suitable for feature extraction". Perhaps an explanation of what you mean with "feature" will make clearer what you mean. Or, is it perhaps the case that your concept of  feature extraction is  the same as template matching? In this case, could your experimental results imply that template matching or feature extraction are not good account of how deep nets perform classification?

---

### Official Review · Reviewer_JQhs · 2024-10-30

**Soundness:** 2
**Presentation:** 2
**Contribution:** 1
**Rating:** 3
**Confidence:** 3

**Summary:**

This paper aims to show why piecewise linear functions realized by ReLU-based convolutional networks perform well on image classification.  The main theoretical result draws a parallel between convolutions and template matching, and shows that for any set of templates that define a visual class, there exists a convnet that has zero error in recognizing the presence of the template embedded in an image.  A toy data study using 2x2 black and white squares and similar template-based study with FashionMNIST on random backgrounds confirm the theoretical work and show the behavior of the same simple template-like convnet setups in an empirical scenario.

**Strengths:**

I agree with much of the initial motivation in the introduction sections, that many theoretical results around the ability of relu networks to realize any piecewise linear function doesn't quite get at why they are effective, particularly since most piecewise linear functions over the space of image matrices are not useful classifiers.

The overall approach of using template matching to bridge between theoretical guarantees and practical instances is promising.

The overall presentation is good, going from intro to the mathematical framework to empirical results, each piece is mostly well explained and connected to each other part of the paper.

**Weaknesses:**

While this presents a nice goal, the current use of template matching only accounts for rigid templates, which I think is overly simplified to provide much significant insight.  ConvNet features can become nonzero (sometimes with different degrees of strength) in the presence of image components with deformations from a single appearance.  [1] shows variation in appearance corresponding to different neurons, for example.  This is important in practical use --- otherwise templates are constrained to exact-match and could be implemented with a single large conv kernel.

Template matching is also used and then layered in older deformable parts models [4], and these are shown in [2, 3] to correspond to shallow convnets.  So the correspondence between image part templates and convnets has been well-established for some time.

The two main theorems in this work are existence proofs, but if any piecewise linear function can be implemented by some network according to the related work in the "Expressiveness" section at l.107, then existence of a network that can implement template matching exactly seems like it would follow from these almost immediately, unless I'm missing something here?  A stronger theoretical result that would be more interesting might relate to numbers of layers, dimensions and/or model capacity to bounds on how well a template function can be approximated, for example.

The empirical results are also rather limited, only studying the responses of a shallow network with single conv layer and fully connected layer.


[1] Zeiler and Fergus ECCV 2014 "Visualizing and Understanding Convolutional Networks"

[2] Girshick et al. CVPR 2015 "Deformable Part Models are Convolutional Neural Networks"

[3] Wan et al. CVPR 2015 "End-to-End Integration of a Convolutional Network, Deformable Parts Model and Non-Maximum Suppression"

[4] Felzenszwalb et al. TPAMI, 2010 "Object detection with discriminatively trained part based models.

**Questions:**

See above.  In addition, there are a few places such as around l.360 that mention a kernel size that grows with the number of layers ("2^i x 2^i kernels").  Is the kernel size actually getting larger in each layer, or is this meant to refer to the receptive field and all kernels are 2x2?

I looked at the proofs in the appendix only skimming, but the one time I did want to read the details, for the proof of Theorem 2, I found a reference to Lemma 4.9 that I can't find.

---

### Official Review · Reviewer_zA5o · 2024-10-30

**Soundness:** 1
**Presentation:** 1
**Contribution:** 1
**Rating:** 1
**Confidence:** 5

**Summary:**

The paper introduces a mathematical framework to analyze the expressiveness of convolutional neural networks (CNNs) in image classification, utilizing piecewise linear functions derived from template matching algorithms. It illustrates the effectiveness of CNNs in feature extraction and explores network interpretability. The study highlights the principle of hierarchical compositionality, demonstrating how simpler features combine to identify complex patterns. Ultimately, it tries to connect theoretical insights with empirical findings, deepening our understanding of CNNs in the realm of computer vision.

**Strengths:**

The paper offers a theoretical framework that enhances the understanding of convolutional neural networks (CNNs) in feature extraction through piecewise linear functions. It addresses interpretability by distinguishing between monosemantic and polysemantic neurons, providing insights into feature recognition. The emphasis on hierarchical compositionality illustrates how CNNs combine features at different levels. Additionally, empirical validation somewhat supports the theoretical claims.

**Weaknesses:**

The paper's reliance on template matching algorithms may not fully capture the complexities of real-world image variations, such as rotations and distortions. The theoretical framework may also lack practical applicability in more complex scenarios. Lastly, the empirical results are based on simplified datasets, which may not generalize to more challenging tasks in computer vision. The paper is more focused on math than experiments, which is a significative drawback. The Fashion-MNIST is a toy dataset that do not ensure us that the experiments generalize for more realistic data. Color dataset should be used. Study Vision Transformers would enhance the value of the paper.

**Questions:**

No questions.

---

### Official Review · Reviewer_8Bwm · 2024-11-04

**Soundness:** 3
**Presentation:** 2
**Contribution:** 2
**Rating:** 5
**Confidence:** 4

**Summary:**

- The draft proposes a mathematical model based on template matching to show how CNNs, which compute piecewise linear functions, excel at feature extraction.
- It proves that simple CNNs can classify images with zero error by detecting features, while deeper networks leverage hierarchical compositionality to combine simpler patterns into complex ones.
- Experimental results on a toy dataset and Fashion-MNIST demonstrate that CNNs achieve high accuracy by capturing feature-specific patterns robust to background (and translation) variations.

**Strengths:**

1. The draft shows piecewise linear functions enable precise, interpretable (because of the templates) feature detection. Based on this, it enhances the theoretical understanding of CNNs. Moreover, the draft also interprets deep CNNs via hierarchical compositionality.
2. Experiments demonstrate the robustness of CNNs to translations and background perturbations, validating the framework’s practical relevance.

**Weaknesses:**

1. It is unclear whether the theoretical validations in the draft bring significant new insights about CNNs (particularly toward practical applications). Some of the existing works provide mathematical explanations for the feature extraction of CNNs (please refer to [A] and [B]).
2. Proofs provided in Appendix B.2 are not easy to follow because of missing explanations/assumptions. They need to be elaborated.
3. Experimentation is very simple (only toy datasets are considered).
4. [Minor] Inconsistent matrix notation (Lines 777 and 785) could be clearer.

Post the discussion phase
- The authors have not provided any response. After revisiting the submission and the comments from fellow reviewers, I keep the original rating.

[A] Mallat, S. (2012). Group invariant scattering. Communications on Pure and Applied Mathematics, 65(10), 1331-1398.
[B] Wiatowski, T., & Bölcskei, H. (2017). A mathematical theory of deep convolutional neural networks for feature extraction. IEEE Transactions on Information Theory, 64(3), 1845-1866.

**Questions:**

1. Does Theorem 1 hold for different kernel sizes? Why the size of kernels is 2 X 2?

---

### Official Review · Reviewer_LsJT · 2024-11-04

**Soundness:** 2
**Presentation:** 1
**Contribution:** 1
**Rating:** 3
**Confidence:** 3

**Summary:**

This paper provides a mathematical framework based on template matching to explain why CNNs with piecewise linear functions work for image classification. The authors provide proof that CNNs can solve template matching problems and demonstrate this on simple synthetic datasets, though their framework doesn't fully explain CNNs' real-world performance.

**Strengths:**

- Attempts to provide mathematical rigor to template matching in CNNs
- Clear experimental setup with reproducible toy examples
- Honest about limitations and gaps in their theory

**Weaknesses:**

- Nothing fundamentally new - repackages known concepts with mathematical formalism
- Experiments are trivially simple (black/white squares and Fashion-MNIST embeddings)
- Theory doesn't scale to real-world challenges with large disconnect between their theoretical construction and actual CNN behavior.

Presentation:
- Poor paper structure, with surprising typos, even in the abstarct (eg. " a fundamental theoretical questions remain answered:" which must have been  " a fundamental theoretical 'question' remain 'unanswered':"
- No dynamic link reference to the Figures. Figure captions are not explanatory.

**Questions:**

See weaknesses.

---

### Note · Authors · 2025-01-01

I have read and agree with the venue's withdrawal policy on behalf of myself and my co-authors.